# Clinical Implementation of Tissue-Sparing Posterior Cervical Fusion: Addressing Market Access Challenges

**DOI:** 10.3390/jpm14080837

**Published:** 2024-08-07

**Authors:** Morgan P. Lorio, Pierce D. Nunley, Joshua E. Heller, Bruce M. McCormack, Kai-Uwe Lewandrowski, Jon E. Block

**Affiliations:** 1Advanced Orthopedics, 499 East Central Parkway, Altamonte Springs, FL 32701, USA; mloriomd@gmail.com; 2Spine Institute of Louisiana, 1500 Line Ave, Ste. 200, Shreveport, LA 71101, USA; 3Department of Neurological Surgery, Thomas Jefferson University, 909 Walnut Street, Philadelphia, PA 19107, USA; 4Department of Neurosurgery, University of California San Francisco, 2320 Sutter Street, Ste. 202, San Francisco, CA 94115, USA; 5Center for Advanced Spine Care of Southern Arizona, Division Personalized Pain Research and Education, Tucson, AZ 85712, USA; 6Independent Consultant, 2210 Jackson Street, Ste. 401, San Francisco, CA 94115, USA

**Keywords:** cervical degenerative disc disease, posterior cervical arthrodesis, clinical outcomes, surgical morbidity, tissue-sparing technique, muscle atrophy, adjacent segment disease, precision-engineered implants and instrumentation, bone graft reduction, reimbursement coding, healthcare policy, CPT codes 22600 and 22840

## Abstract

**Background:** The traditional open midline posterior cervical spine fusion procedure has several shortcomings. It can cause soft tissue damage, muscle atrophy, compromise of the lateral masses and painful prominent posterior cervical instrumentation or spinous process if there is dehiscence of the fascia. Additionally, patients frequently experience the rapid development of adjacent segment disease, which can result in the reemergence of debilitating pain and functional impairment. **Clinical relevance:** Tissue-sparing posterior cervical fusion is an alternative method for treating patients with symptomatic cervical degenerative disc disease. However, widespread clinical adoption has been challenged by ambiguity, misunderstandings and misinterpretations regarding appropriate procedural reimbursement coding. **Technological advancement**: The tissue-sparing posterior cervical fusion procedure was approved by the US Food and Drug Administration (FDA) in 2018 (CORUS™ Spinal System and CAVUX^®^ Facet Fixation System (CORUS/CAVUX); Providence™ Medical Technology). This technique addresses the concerns with traditional spine fusion methods by achieving the stability and outcomes of posterior cervical fusion without the morbidity associated with significant muscle stripping in the traditional approach. This technology uses specialized implants and instrumentation to perform all of the steps required to facilitate bone fusion and provide stability while minimizing tissue disruption. The technique involves extensive bone preparation for fusion and placement of specialized stabilization implants that span the facet joint, promoting natural bone growth and fusion while reducing the need for extensive exposure. This procedure provides an effective, less invasive solution for patients with cervical degenerative disc disease. **Reimbursement and coding clarity:** The article provides a comprehensive rationale for appropriate reimbursement coding for tissue-sparing posterior cervical fusion. This is a critical aspect for the adoption and accessibility of medical technologies. This information is crucial for practitioners and healthcare administrators, ensuring that innovative procedures are accurately coded and reimbursed. **Procedural details and clinical evidence:** By detailing the procedural steps, instruments used and the physiological basis for the procedure, this article serves as a valuable educational resource for spine surgeons and payers to appropriately code for this procedure. **Conclusions:** The description of work for CORUS/CAVUX is equivalent to the current surgical standard of lateral mass screw fixation with decortication and onlay posterior grafting to facilitate posterior fusion. Thus, it is recommended that CPT codes 22600/22840 be used, as they best reflect the surgical approach, instrumentation, decortication, posterior cervical fusion and bone grafting procedures.

## 1. Indications and Problems with Posterior Cervical Arthrodesis

Posterior cervical fusion with bone grafting and temporary external immobilization has been a cornerstone in the treatment of cervical spine conditions since the late 19th century [1]. Over time, the procedure has undergone numerous modifications, particularly in the type and extent of supplementary instrumented fixation, such as the development and refinement of wiring techniques [2,3,4,5,6]. These advancements have contributed to posterior cervical fusion becoming a well-established and widely accepted treatment for cervical degenerative disease [7].

Despite its effectiveness in achieving high fusion rates and satisfactory clinical outcomes, posterior cervical fusion is not without significant peri- and postoperative morbidity [8,9]. One of the primary issues with traditional posterior cervical fusion is the need to strip the paraspinal muscles from the spine and retract these tissues to expose the lateral mass bone. This extensive exposure is associated with several problems, including longer surgery times, increased intraoperative bleeding, risk of infection, extended hospital stays, higher and longer postoperative narcotic pain medication requirements and higher overall complication rates compared with less invasive techniques [10,11,12,13].

The extensive tissue disruption required by traditional posterior cervical fusion can lead to considerable muscle atrophy and compromise neck stability. Additionally, the procedure often necessitates the use of prominent posterior cervical instrumentation, which can be painful for patients and contribute to ongoing discomfort. Fascia may dehisce postoperatively, leading to the splaying of the paraspinal muscles and protruding spinous process. Moreover, patients undergoing traditional posterior cervical fusion to revise an anterior pseudoarthrosis frequently report persistent moderate to severe pain even after achieving solid arthrodesis. Nearly half of these patients experience ongoing pain despite the technical success of the surgery [10]. This indicates that while the fusion may be mechanically successful, it does not always translate into symptomatic relief for the patient.

Given the challenges and shortcoming associated with traditional open midline posterior cervical fusion, there is a clear need for an alternative, less disruptive, tissue-sparing technique.

## 2. Historical Perspective

The unique anatomy of the human spine, including the cervical region, consists of a series of stacked three-joint complexes, which include the intervertebral disc and the two posterior facet joints. The facet joints are located bilaterally on the superior and inferior aspects of the vertebral arches. These joints play a crucial role in guiding and limiting the movement of the spine. The junction of the facet joints demarcates the lateral masses of adjacent vertebrae, forming the functional unit of motion. Arthrodesis can be achieved through fusion of the anterior column, the posterior column (i.e., facet joints) or both. The primary goal of all cervical fusion procedures is to restore anatomical alignment, decompress the neural elements, stabilize the joint to resist forces in all axes of movement and provide the optimal environment for bone fusion to occur. In the absence of adequate bone fusion, all instrumentation constructs will ultimately fail [5].

## 3. The Rationale for Cervical Arthrodesis

Posterior cervical arthrodesis, commonly referred to as spinal fusion, involves the surgical fusion of two or more vertebrae to eliminate painful motion and restore spinal stability. The American Academy of Orthopaedic Surgeons describes this procedure as a “welding process” [14]. The primary objective is to unite the vertebrae so that they heal into a single, solid bone, thereby eliminating intervertebral motion at the site of fusion. Achieving both mechanical and functional stability in the fusion construct is essential [15]. Post-laminectomy kyphosis is another target for posterior cervical fixation and stabilization [16,17].

For successful arthrodesis, the fusion mass must be fully ossified from end to end, consisting of remodeled trabeculated bone without breaks or cracks that could cause micromotion and potentially lead to pseudoarthrosis. To promote bone formation across an anatomical void, biomechanical stabilization and a physiological stimulus are required. Bone grafts, often encased in a fusion device, provide the necessary environment to induce ossification across the joint space, forming an extra-anatomic bony bridge.

Osseointegration is further enhanced by placing the grafting materials on the bleeding surface of freshly decorticated, vascularized bone [6,18,19]. As a result, imaging evaluation criteria focus on the solidity and stability of the developing fusion mass [20,21,22]. These criteria have been adopted by US federal regulators as mandatory elements in the review, evaluation, and approval of spinal fusion systems [23]. Hence, posterior cervical arthrodesis aims to eliminate painful motion and stabilize the spine by fusing vertebrae into a single, solid bone. This procedure relies on the strategic use of bone grafts and the creation of an optimal physiological environment to ensure successful osseointegration and long-term stability.

## 4. The Genesis and Advancement of Tissue-Sparing Posterior Cervical Fusion

The genesis of tissue-sparing posterior cervical fusion can be traced back to the facet distraction technique developed by Goel [24,25,26]. This technique, originally performed for C1-C2 distraction with an open posterior technique for basilar invagination, involves distracting the facet joints to increase foraminal height and volume, thereby allowing for the indirect decompression of the exiting nerve root [27].

Over time, this method has been extensively refined, leading to the development of advanced systems such as the CORUS™ Spinal System and CAVUX^®^ Facet Fixation System (CORUS/CAVUX) (Providence™ Medical Technology, Pleasanton, CA, USA) (Figure 1). CORUS is a set of disposable instruments used to access and prepare the posterior cervical spine for joint fusion by decortication of bone surfaces, including the posterior lateral mass and facet joints, combined with the application of allograft or autograft in patients with or without anterior or posterior instrumentation. CAVUX cages are used with ALLY Bone Screws as part of the CAVUX Facet Fixation System and achieve facet fixation by spanning the interspace with fixation points at each end of the construct.

These modern systems utilize cervical cages combined with bone grafts and integrated bone screws, which are implanted into the rostral and caudal lateral masses from the facet joints through a posterior approach. This approach traverses the same posterior structures as the traditional open procedure and can be achieved by using a single mid-line incision or smaller bilateral incisions, facilitating decortication, preparation and then stabilization across the three-joint complex. During the procedure, the joints are prepared, and the facet and lateral mass are decorticated. The implants are then press-fit into the inter-facet region and anchored into the superior and inferior lateral masses of the joint by using bone screw fixation to promote stabilization and fusion. Bone graft material is placed inside the cage, within the decorticated facet trough and atop the joint space extending to the lateral mass.

In contrast, conventional open posterior cervical fusion entails a midline incision, cutting the ligamentum nuchae and stripping the muscular and ligamentous attachments off the spinous process, lamina and lateral mass in a medial-to-lateral direction. Soft tissue dissection has to be extended at least one spinal level above and below the intended arthrodesis to gain exposure of the lateral border of the spine. The entire dorsal spine is exposed to air, but arthrodesis and instrumentation (lateral mass screws) are typically only performed on the lateral mass and facet. Hence, most of the soft tissue dissection is not necessary for arthrodesis.

The tissue-sparing approach can be performed with this standard open exposure but can also be performed with much less soft tissue dissection to support multilevel anterior fusion. The surgeon incises the skin with a cut as small as a dime and incises the ligamentum nuchae off the spinous process. A facet access tool is inserted through this cut and is guided under fluoroscopy to the facet while splitting and deflecting paraspinal muscles laterally. Soft tissue attachments to the spine are largely preserved. The facet access tool is docked into the facet joint and serves as a post for the placement of a viewing port (DiViNE™ Portal System; Providence™ Medical Technology, Pleasanton, CA, USA) for direct visualization with the naked eye and later a guide tube for decortication. The implant is inserted, and the bone graft is placed.

The relatively large footprint of the CORUS/CAVUX cage devices in relation to the facet surface area, combined with their placement under compression, supports osseointegration across the joint [28]. This technique has been shown to have several advantages over traditional methods, offering enhanced safety and effectiveness for the treatment of cervical radiculopathy.

A robust body of evidence underscores the benefits of this tissue-sparing procedure, marking it as a significant advancement in the field of posterior cervical fusion surgery [9,29,30,31,32,33]. The combination of reduced tissue damage, enhanced stability and improved clinical outcomes positions the CORUS/CAVUX as a sufficient alternative to traditional posterior cervical fusion techniques [34]. The description of the system’s instruments and implants and its FDA-approved indications and contraindications are summarized in Table 1. It should be noted that the FDA recently (18 June 2024) expanded the indications for this system to include adjunctive use with anterior cervical discectomy and fusion (ACDF) for up to three consecutive vertebral levels to form circumferential arthrodesis (CORUS™ PCSS, K241035).

## 5. Reimbursement Status Rationale

The tissue-sparing posterior cervical fusion procedure involves a series of procedural steps that reflect the work necessary to establish arthrodesis across the facet joints. Specifically, specialized instruments are used to access and prepare the posterior cervical spine for joint fusion by decortication of bone surfaces, including the posterior lateral mass and facet joints, combined with the application of allograft or autograft in patients with or without supplementary anterior or posterior instrumentation. After the incision and exposure of the bony elements, seven different instruments are used to provide joint access, bone preparation and bone graft delivery, including a guide tube, an access chisel to create a proper pathway, a trephine decorticator designed for lateral mass decortication, a rasp decorticator used for articular surfaces, a rotary decorticator for articular surfaces and lateral mass, a multi-tool and a bone graft tamp. The cervical cages are then placed bilaterally through a posterior surgical approach and span the interspace with additional points of screw fixation at each end of the construct (Figure 2).

Recently, Hagland et al. [9] reported an approximate 90% fusion rate at two years of follow-up in patients undergoing posterior cervical fusion using CORUS/CAVUX as a salvage operation for failed anterior cervical discectomy and fusion (ACDF). Combining CORUS/CAVUX with ACDF to form circumferential fusion leverages the stability of an anterior interbody implant coupled with inter-facet cages to create three points of fixation for fusion of the anterior and posterior columns [35] (Figure 3).

With all posterior cervical fusion procedures, the bulk of the developing fusion mass is encompassed within the inter-facet region, as compressive forces across this joint space encourage mineral apposition [36]. The bony junction between the superior and inferior articular processes, the so-called lateral mass, is separated medially from the lamina by the medial facet line and is often incorrectly ascribed as the anatomical region of bone union [3]. Ultimately, the facet complex motion segment is fused regardless of the technique. The lateral mass is simply where hardware is placed by using either a Magerl or pedicular trajectory if modern traditional screws are used [37]. Lateral mass fusion as a procedural description is, therefore, a misnomer.

Clearly, Current Procedural Terminology (CPT) code 22600 describes surgical approach, decortication and posterior cervical fusion using bone graft material in line with the procedural description of CORUS/CAVUX as noted above. Specifically, CPT code 22600 stipulates arthrodesis, posterior or posterolateral technique, single interspace; cervical below C2 segment, with or without CPT code 22840, posterior non-segmental instrumentation, one interspace. There is no restriction regarding the invasiveness of the procedure, and importantly, CORUS/CAVUX is primarily used in combination with ACDF and/or additional posterior instrumentation (Figure 3). From this perspective, CORUS/CAVUX meets the spirit of the code.

On the contrary, with the operative effort involved and the occurrence of the fusion of the posterior column, it is evident that this tissue-sparing procedure does not correspond with the description of CPT 0219T. This tracking code was introduced on 1 January 2010, to address the growing number of needle-based interventions being employed by pain physicians and interventionalists to treat neck pain syndromes. In fact, the cluster of codes included with CPT 0219T by the Centers for Medicare & Medicaid Services is described as Facet Joint Interventions for Pain Management and includes mostly injection and ablation interventions intended as brief, outpatient or in-office procedures [38]. CPT code 0219T describes procedures that expand and stabilize the facet joint space and lessen pain due to degenerative changes or trauma in the cervical vertebrae region. For example, facet allografts (e.g., bone dowels) can be introduced into the joint space as a stand-alone procedure for facet pain [39]. Typically, these allografts are used earlier in the continuum of care as an intermediate, conservative measure to address or prevent minor instability, mechanical back pain or degenerative joint disease. They do not precipitate fusion. In a series of 96 patients treated with facet bone dowels, Pirris et al. [40] found that only 6 patients (6.3%) demonstrated evidence of inter-facet fusion on postoperative imaging.

## 6. Market Access and Policy Impact

Ensuring market access for innovative surgical technologies like CORUS/CAVUX is crucial for their widespread adoption and integration into clinical practice. Addressing barriers to market access involves not only demonstrating the clinical efficacy and safety of these technologies but also securing appropriate reimbursement through the establishment of CPT codes. The inclusion of new technologies in healthcare reimbursement systems has broader implications for healthcare policy, influencing how new surgical techniques are adopted and utilized.

Publishing detailed studies and articles on these advanced surgical techniques can significantly impact policy decisions. Such publications provide evidence-based support for the efficacy and benefits of these technologies, making a compelling case for their inclusion in standard practice. By influencing policy decisions, these articles can facilitate the adoption of these advanced techniques, ensuring that they are accessible to a broader range of patients who can benefit from improved surgical outcomes and reduced recovery times.

## 7. Interdisciplinary Appeal and Broader Implications

The content of this research is not only relevant to spine surgeons but also to healthcare policymakers, medical coders and healthcare administrators. By addressing the interdisciplinary nature of market access and policy impact, this research broadens its readership and influence. Healthcare policymakers can use the evidence presented to make informed decisions about which technologies to support and integrate into healthcare systems. Medical coders play a crucial role in the reimbursement process, and understanding the nuances of new CPT codes ensures accurate billing and compensation for these advanced procedures. Healthcare administrators, who manage the financial and operational aspects of medical institutions, can also benefit from understanding the cost implications and potential savings associated with these new technologies.

The establishment of appropriate CPT codes is a critical step in overcoming market access barriers [41,42,43]. For instance, categorizing CORUS/CAVUX as Category I fusion underscores its established efficacy and acceptance in the medical community. This distinction can significantly influence patient access to FDA-approved technologies in various healthcare settings.

## 8. Influencing Healthcare Policy and Practice

By providing a comprehensive rationale for appropriate reimbursement coding, this research supports equitable access to innovative surgical techniques. Failure to appropriately reimburse these technologies could limit their adoption, thereby restricting patient access to potentially life-changing treatments. Ensuring that these procedures are accurately coded and reimbursed is essential for their integration into routine clinical practice.

The broader implications of this research extend to healthcare policy and the adoption of new technologies. Policymakers and healthcare providers must work together to ensure that advancements in surgical techniques are supported by robust evidence and are financially accessible. This collaborative effort will help integrate cutting-edge surgical innovations into clinical practice, ultimately improving patient outcomes and advancing the field of spinal surgery.

## 9. Discussion

The procedural evolution of posterior cervical fusion has followed a typical developmental path over time from more invasive to less invasive as a direct result of the need to reduce surgical morbidity, speed up recovery and improve patient outcomes. In fact, the CORUS Spinal System received regulatory clearance from the Food and Drug Administration (FDA) initially in 2018, specifically indicated to be used to perform posterior cervical fusion in patients with cervical degenerative disc disease (C3-C7). The objectives of this perspective article are to articulate a barrier to market access for CORUS/CAVUX for use in posterior cervical spine fusion and recommend appropriate reimbursement coding. We provide this technology as a case where the refinement of a surgical technique that accomplishes similar outcomes with markedly less morbidity can be hindered by a reimbursement system that is slow to adapt to procedural advancements.

From a reimbursement standpoint, there is a small minority that question whether this tissue-sparing fusion procedure should be classified under current CPT Category I codes (CPT 22600 and 22840) or the Category III tracking code (CPT 0219T). This distinction has important ramifications regarding the choice of appropriate treatment for a patient, as reimbursement for procedures utilizing a Category III tracking code is invariably determined on a case-by-case basis with prior authorization. There are no assigned fees for these Category III codes, and coverage is often limited or nil [44].

## 10. Conclusions

The utilization of tissue-sparing posterior cervical fusion promises enhanced stability and clinical outcomes with reduced morbidity. These advancements, supported by robust clinical evidence, demonstrate the potential to substantially improve the treatment of debilitating cervical degenerative disc disease. Ensuring market access for these innovative technologies is crucial for their adoption and integration into clinical practice. Using the case of CORUS/CAVUX as an illustrative example, it is evident that addressing barriers, such as securing appropriate reimbursement codes, has broader implications for healthcare policy and the widespread utilization of advanced surgical techniques. Despite FDA approval for posterior cervical fusion in patients with cervical degenerative disc disease and historical recognition as an arthrodesis procedure, CORUS/CAVUX has faced some unnecessary headwinds in gaining market access in the US due, in part, to ambiguity regarding appropriate procedural coding. Notwithstanding, an independent, certified coding organization has confirmed the applicability of current posterior cervical fusion codes (CPT 22600 and 22840) for this procedure [45].

Detailed studies and publications play a vital role in influencing policy decisions and facilitating the adoption of these technologies, ensuring that patients benefit from improved surgical outcomes and reduced recovery times. The interdisciplinary appeal of this research extends its relevance beyond spine surgeons to include healthcare policymakers, medical coders and administrators. By providing a comprehensive rationale for appropriate reimbursement coding, the authors’ research supports equitable access to innovative surgical techniques. Accurate coding and reimbursement are essential to integrating these procedures into routine practice, ultimately advancing surgical techniques and improving patient outcomes.

## Figures and Tables

**Figure 1 jpm-14-00837-f001:**
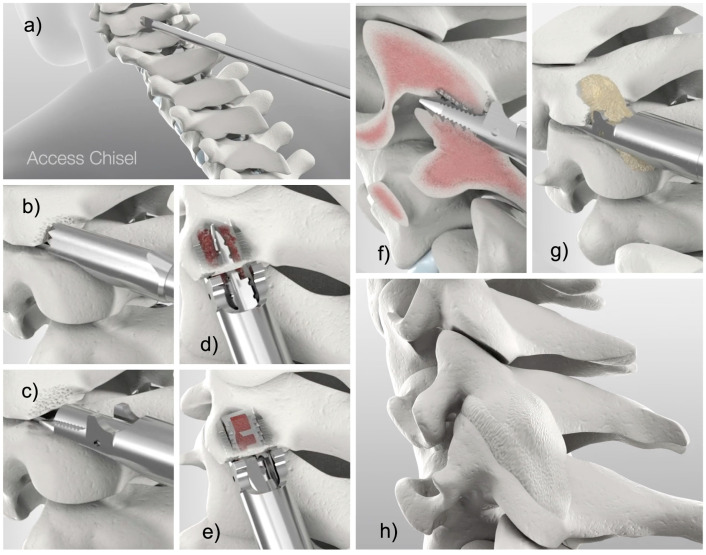
Shown are the CORUS/CAVUX surgical steps: (**a**) minimally invasive posterior access to the cervical spine docking the access chisel at the facet joint complex, (**b**) decortication of the lateral mass with the trephine decorticator, (**c**) removal of the facet capsule and decortication of the articular surfaces with the rasp decorticator, (**d**) application of the rotatory decorticator, (**e**) delivery of the CAVUX cage, (**f**) delivery of the ALLY^®^ bone screw, (**g**) bone graft placement with the bone graft applicator and (**h**) oblique view of the posterior cervical spine with bridging bone indicating successful fusion.

**Figure 2 jpm-14-00837-f002:**
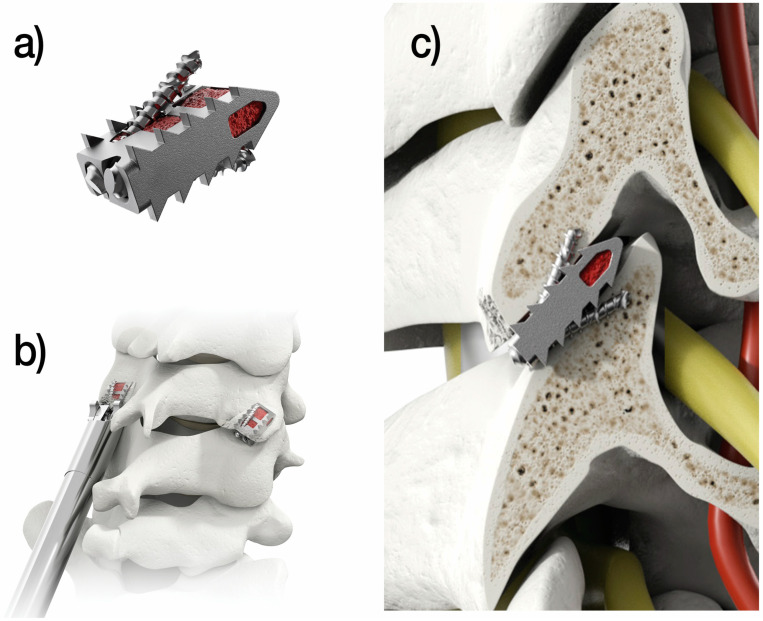
The tissue-sparing posterior cervical fusion procedure involves the implantation of the (**a**) CORUS™ PCSS, an integrated construct comprising a cage and two fixation screws, (**b**) placed bilaterally through a posterior surgical approach, (**c**) spanning the interspace and including additional screw fixation points at each end of the construct to provide trans-facet stabilization.

**Figure 3 jpm-14-00837-f003:**
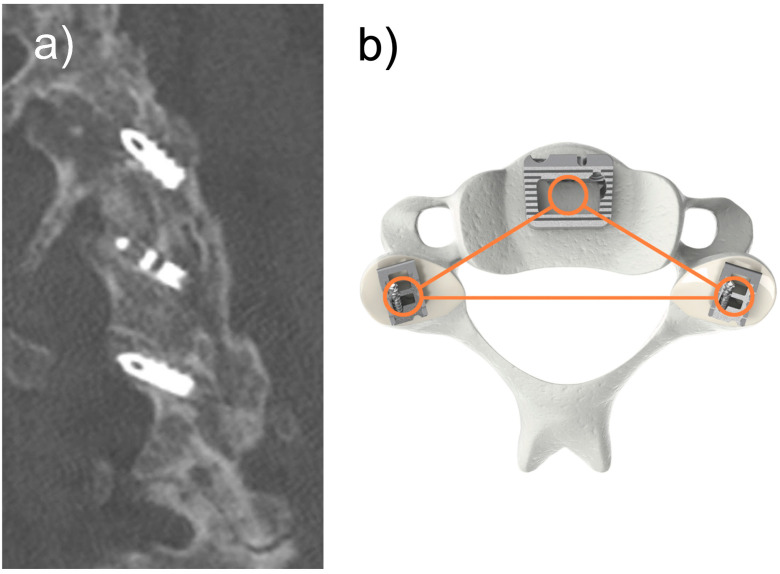
Shown are (**a**) the sagittal view two years after 3-level posterior cervical fusion demonstrating abundant ossification and bridging trabecular bone across the C3/4, C4/5 and C5/6 facet joints confirmed by multiplanar computed tomography scan and (**b**) a tri-force of fixation and support that leverages the stability of the anterior implant coupled with inter-facet cages to create three points of fixation for fusion of the anterior and posterior columns.

**Table 1 jpm-14-00837-t001:** Indications for the use of the CORUS™ Spinal System family of implants for posterior cervical fusion.

Device Name	General Description and FDA Indications for Use
CORUS Spinal System	The CORUS™ Spinal System-X is a set of instruments indicated to be used to perform posterior cervical fusion in patients with cervical degenerative disc disease.
CORUS Posterior Cervical Stabilization System (PCSS)	CORUS Posterior Cervical Stabilization System (PCSS) is posterior spinal instrumentation with integrated screw fixation intended to provide immobilization and stabilization of spinal segments. CORUS PCSS is placed through a posterior surgical approach in up to 3 consecutive levels of the cervical spine (C3-C7) and achieves bilateral facet fixation by spanning the facet interspace at each level with points of fixation at each end of the construct. CORUS PCSS is intended as an adjunct to posterior cervical fusion (PCF) and is only intended to be used in combination with anterior cervical discectomy and fusion (ACDF) at the same level(s). CORUS PCSS is indicated for skeletally mature patients with degenerative disc disease (DDD). DDD is defined as radiculopathy and/or myelopathy, neck and/or arm pain of discogenic origin as confirmed by radiographic studies. CORUS PCSS is to be used with autogenous bone and/or allogenic bone graft.
CAVUX Facet Fixation System (FFS)	The CAVUX Facet Fixation System (CAVUX FFS) is indicated for patients needing revision for anterior pseudarthrosis at one level (C3 to C7), with autogenous and/or allogenic bone graft. It consists of an integrated construct consisting of a CAVUX Cage and a single ALLY Bone Screw. It is placed bilaterally through a posterior approach and spans the interspace with fixation points at each end. CAVUX FFS is intended for temporary stabilization as an adjunct to posterior cervical fusion in skeletally mature patients.

## Data Availability

Not applicable.

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
