# Peer review of "Clinical Implementation of Tissue-Sparing Posterior Cervical Fusion: Addressing Market Access Challenges"

_jpm, 2024, doi:10.3390/jpm14080837_

Round 1
Reviewer 1 Report
Comments and Suggestions for Authors
Congratulations to the authors.This is an important work. Personally I think there is place for some clarifications.
1.I would like to have more clarification how the surgical approach is considered tissues sparing! You have to expose the joint and to damage some tissue!
2. is the technique considered minimal invasive or less invasive surgery?
3. Can the authors explain why despite being many years on the market the system is used less and they are not many papers or research!
4. Is there any chance to cause kyphosis!
Author Response
Reviewer #1
Congratulations to the authors. This is an important work. Personally, I think there is place for some clarifications.
1. I would like to have more clarification how the surgical approach is considered tissues sparing! You have to expose the joint and to damage some tissue!
We have provided additional descriptive text in section 4 (lines 163-180) differentiating the standard open posterior cervical fusion procedure and the tissue-sparing procedure.
2. Is the technique considered minimal invasive or less invasive surgery?
As we note in the expanded section 4, the tissue-sparing procedure is minimally-invasive with respect to the “dime size” entry point and the relatively limited dissection necessary.
3. Can the authors explain why despite being many years on the market the system is used less and they are not many papers or research!
The tissue-sparing cervical fusion and instrumentation technology has been studied broadly and there are over 30 peer-reviewed journal publications demonstrating robust fusion results, biomechanical stability on par with lateral mass screws, and excellent patient outcomes. We have expanded the bibliography of supporting literature in section 4 (line 188) to include additional citations.
Additionally, the FDA has also recently completed analysis on a 227-patient prospective, multi-center, randomized, controlled clinical trial which resulted in demonstrating superiority of circumferential fusion (ACDF + CORUS PCSS) over ACDF alone, and was the basis for the most recent FDA clearance for 3-level cervical fusion (date: June 18, 2024). The primary results of this trial are under review at another journal. We have noted this regulatory approval in the text in section 4 (lines 192-195).
4. Is there any chance to cause kyphosis!
The risk of kyphosis is minimal as this procedure is most often performed in conjunction with ACDF to form a stable circumferential fusion or as a salvage procedure for a failed ACDF where there is already anterior column reconstruction.
Reviewer 2 Report
Comments and Suggestions for Authors
Dear Authors,
Thank you for allowing me to review this article. It provides a clear perspective of the impact that a novel, tissue-sparing, posterior spinal fusion technique has on the surgical treatment of cervical degenerative disease. The article is well structured; it incorporates not only a surgical-centered approach but also a market-related evaluation. The technical details are well presented. I appreciated very much the areas related to reimbursement and coding issues, which we do not see frequently mentioned in a paper. I would have appreciated it if you could provide more references related to the use of the new spinal fusion technique in practice. Otherwise, the article looks great and is good to be published!
Author Response
Reviewer #2
Dear Authors,
Thank you for allowing me to review this article. It provides a clear perspective of the impact that a novel, tissue-sparing, posterior spinal fusion technique has on the surgical treatment of cervical degenerative disease. The article is well structured; it incorporates not only a surgical-centered approach but also a market-related evaluation. The technical details are well presented. I appreciated very much the areas related to reimbursement and coding issues, which we do not see frequently mentioned in a paper. I would have appreciated it if you could provide more references related to the use of the new spinal fusion technique in practice. Otherwise, the article looks great and is good to be published!
We appreciate the kind words of the reviewer. We have expanded the bibliography of supporting literature in section 4 (line 188) to include additional citations supporting the tissue-sparing cervical fusion procedure.